# Downregulation of 5-hydroxymethylcytosine is associated with the progression of cervical intraepithelial neoplasia

**Masaya Kato**[1,2], **Ichiro Onoyama**[1]*, **Minoru Kawakami**[1], **Sachiko Yoshida**[1], **Keiko Kawamura**[1], **Keisuke Kodama**[1], **Emiko Hori**[1], **Lin Cui**[1], **Yumiko Matsumura**[1], **Hiroshi Yagi**[1], **Kazuo Asanoma**[1], **Hideaki Yahata**[1], **Atsuo Itakura**[2], **Satoru Takeda**[2], **Kiyoko Kato**[1]

**1** Department of Obstetrics and Gynecology, School of Medical Sciences, Kyushu University, Fukuoka, Japan, **2** Department of Obstetrics and Gynecology, School of Medical Sciences, Juntendo University, Tokyo, Japan

\* ichirou@med.kyushu-u.ac.jp

**Data Availability Statement:** All relevant data are within the paper and its Supporting Information files.

## Abstract

Around the world, cervical cancer is one of the most common neoplastic diseases among women, and the prognosis of patients in an advanced stage remains poor. To reduce the mortality rate of cervical cancer, early diagnosis and treatment are essential. DNA methylation is an important aspect of gene regulation, and aberrant DNA methylation contributes to carcinogenesis and cancer progression in various cancers. Although 5-methylcytosine (5mC) has been analyzed intensively, the function of 5-hydroxymethylcytosine (5hmC) has not been clarified. The purpose of our study was to identify the molecular biomarkers for early diagnosis of cervical tumors due to epigenetic alterations. To assess the clinical relevance of DNA methylation, we used immunohistochemistry (IHC) to characterize the level of 5hmC in 102 archived human cervical intraepithelial neoplasia (CIN) samples and cervical cancer specimens. The level of 5hmC was significantly decreased between CIN2 and CIN3. The progression of cervical tumors is caused by a reduction of TP53 and RB1 because of HPV infection. We observed that *Tp53* and *Rb1* were knocked down in mouse embryonic fibroblasts (MEF), a model of normal cells. The level of 5hmC was reduced in *Tp53*-knockdown cells, and the expression levels of DNA methyltransferase 1 (DNMT1) and ten-eleven translocation methylcytosine dioxygenase 1 (TET1) were induced. In contrast, there was no significant change in *Rb1*-knockdown cells. Mechanistically, we focused on apolipoprotein B mRNA editing enzyme, catalytic polypeptide-like (APOBEC) 3B (A3B) as a cause of 5hmC reduction after *TP53* knockdown. In the human cell line HHUA with a wild-type *TP53* gene, A3B was induced in *TP53*-knockdown cells, and A3B knockdown recovered 5hmC levels in *TP53*-knockdown cells. These data indicate that TP53 suppression leads to 5hmC reduction in part through A3B induction. Moreover, IHC showed that expression levels of A3B in CIN3 were significantly higher than those in both normal epithelium and in CIN2. In conclusion, 5hmC levels are decreased between CIN2 and CIN3 through the TP53-A3B pathway. Since A3B could impair genome stability, 5hmC loss might increase the chances

**Funding:** This work was supported by the Japan Society for the Promotion of Science (JSPS) KAKENHI grants JP20K09646 (IO) and JP19H03800 (KK).

**Competing interests:** The authors have declared that no competing interests exist.

of accumulating mutations and of progressing from CIN3 to cervical cancer. Thus, these epigenetic changes could predict whether CINs are progressing to cancer or disappearing.

## Introduction

Cervical cancer is one of the most common cancers among women globally. Although surgical techniques have advanced, and systemic therapies have improved, the efficacy of treatment and prognosis for patients with advanced cervical cancer remain poor [1, 2]. Cervical squamous intraepithelial neoplasia (CIN) is a pre-malignant lesion of cervical cancer. Atypical squamous alterations occur sequentially, ranging from mild, to moderate, to severe (CIN 1, 2, 3). To reduce the mortality rate of cervical cancer, early diagnosis and treatment are essential. However, it is sometimes difficult to categorize those cases with progressing CIN. Hence, frequent follow-up examinations are necessary to detect progressing CIN [3–5]. Accordingly, it is particularly important to discover new molecular biomarkers to distinguish CIN stages prior to their development into cancer.

Epigenetics is a promising and expanding field in pathologic studies. DNA methylation, including 5-methylcytosine (5mC) and 5-hydroxymethylcytosine (5hmC), is known to be involved in tumorigenesis. 5mC is supposed to be a relatively stable DNA modification and methylation of the promoter inhibits transcription of genes [6]. Cytosine methylation is catalyzed by DNA methyltransferases (DNMTs) [7]. On the other hand, 5hmC is an intermediate product in DNA demethylation. The ten-eleven translocation (TET) protein family (TET1, TET2, TET3) catalyzes the conversion of 5mC to 5hmC, 5-formylcytosine, and 5-carboxylcytosine, leading to DNA demethylation [8, 9].

5hmC levels are decreased in various malignant tumors [10–12]. In cervical cancer, the expression of 5hmC was significantly decreased in cervical squamous cell carcinoma compared with normal cervical tissues [13]. However, sequential change of 5hmC levels in cervical tumors, including CIN1, CIN2, CIN3 and cervical cancers have not been investigated. Given that epigenetic changes in precancerous lesions could lead to malignant transformation, monitoring 5hmC levels might be a sensitive method to identify those CIN patients highly susceptible to the development of cervical cancers. Thus, we analyzed the epigenetic characteristics of cervical tumors, focusing on 5hmC and 5mC changes. Here, we showed that 5hmC levels decrease with the progression of CIN subtypes to cervical cancer, and that 5hmC downregulation is mediated in part by the TP53-A3B pathway.

## Materials & methods

### Cervical intraepithelial neoplasia and cervical cancer tissues

Cervical tumor specimens from 102 female patients who were treated at Kyushu University Hospital between 2013 and 2018 were included. Tumors were histologically characterized as CIN1, CIN2, CIN3, or cervical cancer. The median age of the patients was 46 years old. Informed consent was obtained from all patients prior to enrollment in the study. The ethics committee of Kyushu University Graduate School approved the study protocol.

### Immunohistochemistry

4-μm thick sections were cut from formalin-fixed and paraffin-embedded blocks. The sections were deparaffinized and rehydrated. For antigen retrieval, the sections were boiled in Reveal

Decloaker RTU at 80˚C for 1 h. For detection of 5hmC and 5mC, the slides were incubated in 2N HCl at room temperature for 15 min. The slides were incubated in primary antibody (S1 Table) at 4˚C overnight, and incubated in the secondary antibody labeled with Alexa Fluor (Invitrogen) at RT for 1 h. The slides were sealed in ProLong Gold Antifade Mount with DAPI (Invitrogen). The signals were detected with a confocal laser microscope (Nikon). For A3B, endogenous peroxidases were blocked using 3% hydrogen peroxide in methyl alcohol for 15 min. The slides were incubated with a primary antibody at 4˚C overnight and with a secondary antibody at room temperature for 1 h. The IHC signal was detected using DAB (Wako), and the nuclei were stained with hematoxylin as we have described previously [14]. In each case, the total IHC score was calculated by at least 2 examiners using a BZ-X710 microscope (Keyence). The total score was based on the sum of the score for the proportion of positive cells and a score for staining intensity (S2 Table).

## Cell culture

Mouse embryonic fibroblasts (MEFs) were isolated from E12.5 C57/BL6 embryos. HEK293T cells were isolated from human embryonic kidneys expressing abundant SV40 large T antigen. HHUA cells, a human endometrial cancer cell line with wild-type TP53, were purchased from RIKEN Bioresource Center. These cells were maintained according to the provider's instructions.

## Plasmids

Lentiviral miR-E-based expression vectors (pLKO1) were purchased from Addgene. Two shRNA hairpins were used in all experiments and representative ones are reported. The target sequences of shRNA hairpins are listed in S3 Table.

## Transfection and lentivirus packaging

Lentivirus for infection was packaged in HEK293T cells. HEK293T cells were transfected with the lentiviral vectors using Effectene (QIAGEN). Forty-eight h after transfection, the virus-containing culture supernatant was collected and passed through a 0.45 μm filter. The virus-containing supernatant was added to cells in the presence of polybrene at a final concentration of 8 ng/μL.

## Real time quantitative reverse transcription PCR (qRT-PCR)

Total RNA from cells was extracted using ISOGEN (NIPPON GENE) following the manufacturer's instructions and the RNA was diluted to 250 ng/μL. The cDNA was synthesized with ReverTra Ace (TOYOBO). qRT-PCR was performed using SYBR Premix Ex Taq (Takara Bio) and the CFX Connect Real-time system (Bio-Rad). Each reaction was carried out on a minimum of 3 biological replicates, and the PCR conditions were described previously [15]. The expression of mRNA is presented as the relative copy number normalized to the housekeeping gene *RSP29* and/or *Oaz1* in mice [16]. The PCR primer sequences are listed in S4 Table.

## DNA extraction and ELISA-based quantification of 5hmC and 5mC

Genomic DNA was extracted from cultivated cells. These cells were treated with tail prep buffer containing Proteinase K at 50˚C overnight and DNA was extracted using standard phenol/chloroform methods. Colorimetric quantification of 5mC and 5hmC was performed by using a 5-mC DNA ELISA Kit or a Quest 5-hmC™ DNA ELISA Kit (Zymo Research) according to the manufacturer's instructions.

## Protein extraction and Western blotting

The cells were washed in ice cold PBS and lysed using RIPA buffer. Proteins were run on polyacrylamide gels of optimal concentration at 100 volts in running buffer. The separated proteins on the gels were transferred to nitrocellulose membranes in transfer buffer at 100 volts for 1 h or at 20 volts overnight. Membranes were blocked in 5% skim milk/TBST for 20 minutes and incubated overnight at 4˚C with primary antibodies (S1 Table). The membranes were incubated with secondary antibody at room temperature for 1 h. Specific protein bands were detected using the SuperSignal West Dura Chemiluminescent Substrate (Thermo Fisher Scientific).

## Statistical analysis

To analyze the differences between two groups, either Student's t-test or $\chi^2$ analysis were used. To compare the means of more than two groups, ANOVA was employed. When the results of ANOVA were significant, the Tukey-Kramer method was used. Statistical analysis was performed using Prizm. P values less than 0.05 were considered statistically significant. Each P-value in the figure is indicated as follows: $^*P < 0.05$, $^{**}P < 0.01$, $^{***}P < 0.005$, $^{****}P < 0.001$.

# Results

## 5hmC levels decreased in CIN3 and cervical cancer cases

To determine the clinical relevance of 5hmC in cervical tissues, we used immunochemistry to investigate their levels in a total of 102 archived cases of human cervical cancer and CIN specimens (CIN1: n = 13; CIN2: n = 24; CIN3: n = 26; cervical cancer: n = 39). Normal cervical epithelium was also examined (n = 42) and compared with neoplastic lesions. 5hmC levels were high in normal cervical epithelium. CIN1 and CIN2 showed 5hmC levels comparable to normal epithelium. However, CIN3 and cervical cancers showed significantly lower levels of 5hmC (Fig 1A and 1B, Table 1). Similarly, 5mC levels were also significantly reduced in CIN3 and cancer (S1 Fig, S5 Table, Normal: n = 45; CIN1: n = 10; CIN2: n = 20; CIN3: n = 27; cervical cancer: n = 38). These results indicated that DNA methylation status changed markedly between CIN2 and CIN3 and that epigenetically CIN3 more closely resembles carcinoma than either CIN1 or 2. Thus, measurement of 5hmC level could be useful to distinguish between them.

## Tp53 suppression reduced the level of 5hmC, whereas it induced DNMT1 and TET1

The most common cause of cervical cancer is human papilloma virus (HPV) infection. Viral oncoproteins E6 and E7 suppress Tp53 and Rb1, respectively [17, 18]. To examine whether the functions of E6 and E7 were involved in the reduction of 5hmC and 5mC, we used mouse embryonic fibroblasts (MEFs) as a model of normal cells because they are primary culture cells lacking *Tp53* and *Rb1* mutations. First, we knocked down *Tp53* in MEFs by using the lentivirus vector system (Fig 2A) and examined 5hmC levels. Knockdown efficiency was confirmed by real time PCR, and 5hmC levels were measured with 5hmC ELISA. In *Tp53* knockdown cells, the level of 5hmC was significantly decreased compared to non-silenced (NS) control cells. Next, we knocked down *Rb1*, and measured 5hmC levels. Contrary to *Tp53* knockdown, the levels of 5hmC were not changed in R*b1*-knockdown cells (Fig 2B). Using 5mC ELISA to analyze these cells, we found that the 5mC level in *Tp53*-knockdown cells (sh-Tp53_2) was significantly lower than in NS control cells. Other *Tp53*-knockdowned cells (sh-Tp53_1) also showed lower levels of 5mC, although the differences between the NS controls were not

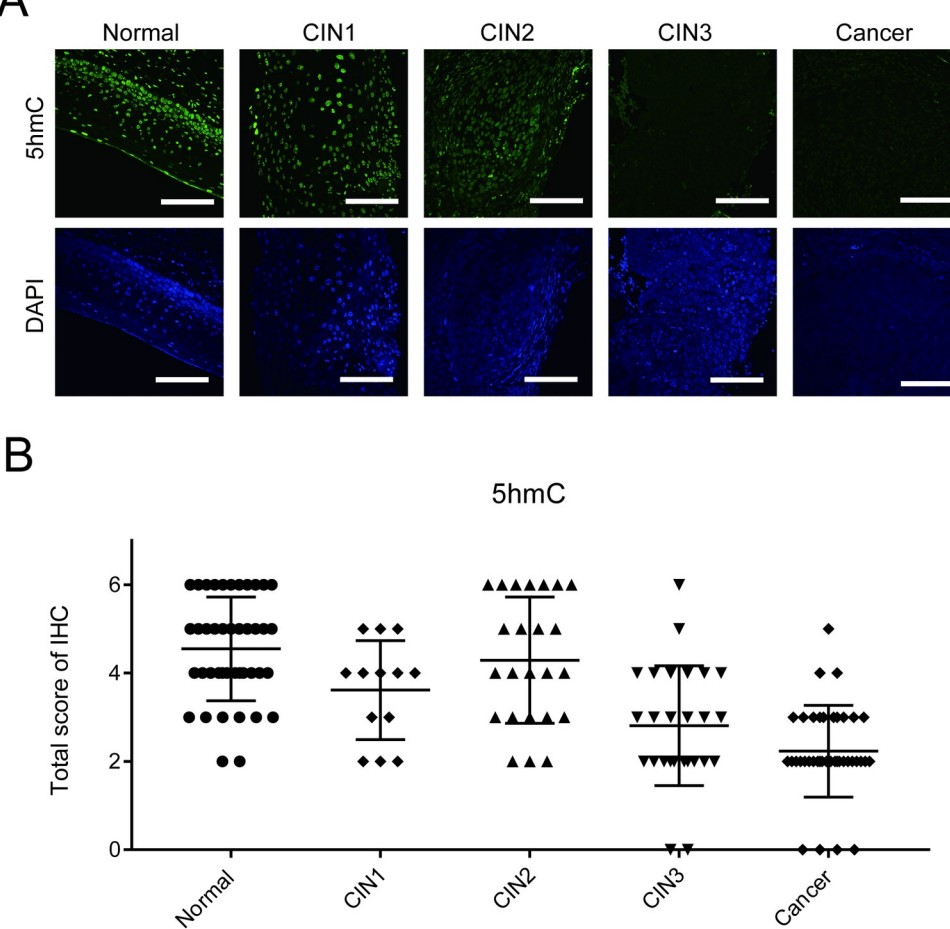

**Fig 1. The levels of 5hmC and 5mC were reduced when CIN2 progressed to CIN3 in cervical tumors.** (A) Representative images of IHC staining of 5hmC and DAPI. Scale bar, 100 μm. (B) Dot plot shows the IHC scores for the levels of 5hmC in normal cervical epithelium, CIN1, CIN2, CIN3 and cancer. Error bars indicate the standard deviation.

**Table 1. Statistical analysis of 5hmC levels in normal cervical epithelia and cervical tumors.** Tukey's multiple comparisons test was performed for the IHC score of 5hmC.

| Tukey's multiple comparisons test | Mean Diff. | 95.00% CI of diff. | Summary | Adjusted P Value |
|---|---|---|---|---|
| Normal vs. CIN1 | 0.9322 | -0.1341 to 1.999 | Ns | 0.1171 |
| Normal vs. CIN2 | 0.256 | -0.6038 to 1.116 | Ns | 0.9232 |
| Normal vs. CIN3 | 1.74 | 0.9015 to 2.578 | **** | <0.0001 |
| Normal vs. Cancer | 2.317 | 1.57 to 3.064 | **** | <0.0001 |
| CIN1 vs. CIN2 | -0.6763 | -1.833 to 0.4807 | Ns | 0.4901 |
| CIN1 vs. CIN3 | 0.8077 | -0.3336 to 1.949 | Ns | 0.2931 |
| CIN1 vs. Cancer | 1.385 | 0.3086 to 2.461 | ** | 0.0046 |
| CIN2 vs. CIN3 | 1.484 | 0.5329 to 2.435 | *** | 0.0003 |
| CIN2 vs. Cancer | 2.061 | 1.189 to 2.933 | **** | <0.0001 |
| CIN3 vs. Cancer | 0.5769 | -0.2737 to 1.428 | Ns | 0.3359 |

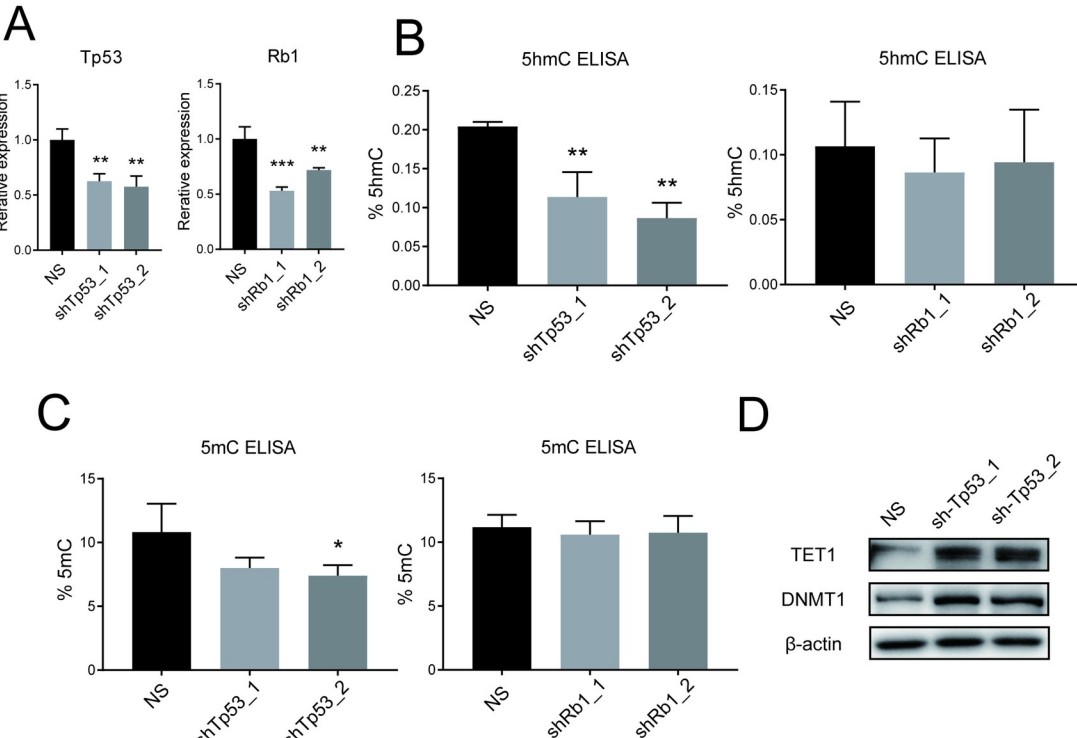

**Fig 2. *Tp53* suppression reduced the level of 5hmC, whereas it induced DNMT1 and TET1.** (A) Bar graph shows the relative mRNA expression levels of *Tp53* and *Rb1*. Knockdown of *Tp53* and *Rb1* were confirmed compared to NS control. (B) Bar graph shows the levels of 5hmC in *Tp53*-knockdown and *Rb1*-knockdown cells using ELISA compared to NS control cells. (C) Bar graph shows the levels of 5mC in *Tp53*-knockdown and *Rb1*-knockdown cells using ELISA compared to NS control cells. (D) Western blot analysis of TET1 and DNMT in *Tp53*-knockdown and NS cells. β-actin was the loading control. Data are representative of at least 3 independent experiments. Error bars indicate the standard deviation. *, P<0.05; **, P<0.01.

significant, perhaps because of *Tp53* knockdown efficiency. On the other hand, 5mC levels were not changed after *Rb1* knockdown (Fig 2C). These data indicated that *Tp53* suppression is involved in 5hmC downregulation and 5mC demethylation in non-transformed cells.

These epigenetic changes encouraged us to examine the molecular mechanisms underlying 5hmC regulation in association with *Tp53* suppression. Toward this end, we examined the expression of DNMT1 and TET1 in *Tp53*-knockdown cells as they regulate DNA methylation and demethylation (Fig 2D). Interestingly, DNMT1 and TET1 proteins were induced in the *Tp53*-knockdown cells compared to the NS control cells. However, the levels of 5hmC and 5mC were both reduced, although DNMT1 and TET1 generally increased 5mC and 5hmC levels. Thus, DNA methylation status is dramatically changed after *Tp53* suppression in non-transformed cells.

## APOBEC3B was induced in cells transitioning from CIN2 to CIN3 in cervical tumors

Given that DNMT1 and TET1 were induced after *Tp53* knockdown, we tried to identify other factors that decreased 5hmC levels after *Tp53* suppression. Thus, we focused on APOBEC3B (A3B). Notably, A3B is upregulated, and its preferred target sequence is frequently mutated and clustered in cervical cancer [19]. Moreover, A3B is positively related to E6, HPV protein, and it is induced by TP53 loss [20–22], and tumor-associated TP53 mutants can promote A3B expression [23]. APOBEC is a DNA deaminase, selectively targeting cytidine on single-

stranded DNA, and it acts as an inhibitor of retrovirus replication [24]. The cytidine deaminase pattern is widespread in human cancers [25, 26]. As mouse cells have the same DNA deaminase coded by *Apobec3*, we examined APOBEC3 protein expression in *Tp53*-knockdown MEFs by Western blotting analysis. APOBEC3 was highly expressed in *Tp53*-knockdown MEFs compared to NS control cells (Fig 3A).

Thus, we examined A3B expression in the human cell line, HHUA, which carries the wildtype *TP53* gene after it has been knocked down. Again, A3B was highly induced in *TP53*-knockdown cells compared to NS control cells (Fig 3B). Then, we investigated whether A3B expression was involved in 5hmC downregulation induced by TP53 knockdown. 5hmC levels were downregulated in HHUA cells with *TP53*-knockdown compared to those in NS cells. Notably, 5hmC levels were recovered in *TP53* knockdown cells when A3B was also silenced (Fig 3C). These data indicate that A3B induced by TP53 knockdown is, at least, involved in 5hmC downregulation, and TP53 loss might reduce 5hmC levels through A3B induction.

These results encouraged us to use IHC to examine the protein levels of A3B in normal cervical epithelium specimens, CIN2 and CIN3. The A3B IHC signals were scored according to the population of positive cells and signal intensities, and the total IHC score was used to evaluate the cases (Normal: n = 30; CIN2: n = 16; CIN3: n = 20). Representative images of tissues are shown in Fig 3B. Higher expression of A3B was observed in the CIN3 cases than in normal cervical epithelium and CIN2 cases (Fig 3C). These results indicated that higher expression of A3B might be associated with progression from CIN2 to CIN3.

## Discussion

We demonstrated that the level of 5hmC significantly decreased during the progression from CIN2 to CIN3. Although TP53 and RB suppression is important for cervical tumor progression, the loss of 5hmC was observed in only *Tp53*-knockdown cells. A3B was induced during the change from CIN2 to 3, and it might be related to cancer promotion and the reduction of 5hmC.

In our study, TET1 and DNMT1 were induced in *Tp53*-knockdown cells. Although their function is the generation of 5hmC and 5mC, the levels of DNA methylation were reduced in these cells. The loss of TP53 is associated with a low level of DNA methylation. Moreover, a high level of DNMT1 expression induces site-specific methylation of promoter regions, which constitute very small parts in the whole genome [27, 28]. Notably, the degradation of TP53 alters epigenetic patterns [29–31]. The specificity of DNMT1 might account for the low level of DNA methylation despite DNMT1 induction. TETs promote DNA demethylation from 5mC to cytosine through 5hmC, 5-formylcytosine and 5-carboxylcytosine. When we knocked down TP53, the DNA methylation level was already low (Fig 2C). From these data, we hypothesize that DNA demethylation after TP53 knockdown might proceed and be completed very quickly. In this case, we would not be able to detect 5hmC upregulation after DNA demethylation was completed.

5hmC loss occurs during cancer progression, and cell proliferation is incompatible with normal levels of 5hmC [32, 33]. However, the critical mechanism underlying 5hmC loss is unclear. Many reports have shown a relationship between 5hmC and genome stability, as 5hmC reduces mutation frequency by marking sites of DNA damage [34, 35]. Moreover, A3B could impair genome stability, which is induced by TP53 loss [22, 36, 37]. These reports support the probability that 5hmC loss is positively related to multiple mutations induced by Tp53 reduction. In our study, the level of 5hmC was markedly different in CIN2 and CIN3. The difference might be caused by mutation frequency.

The strategy for treating CIN3 is surgical resection because of the high risk of tumor progression, although the therapeutic strategy for CIN2 is controversial in some countries [38].

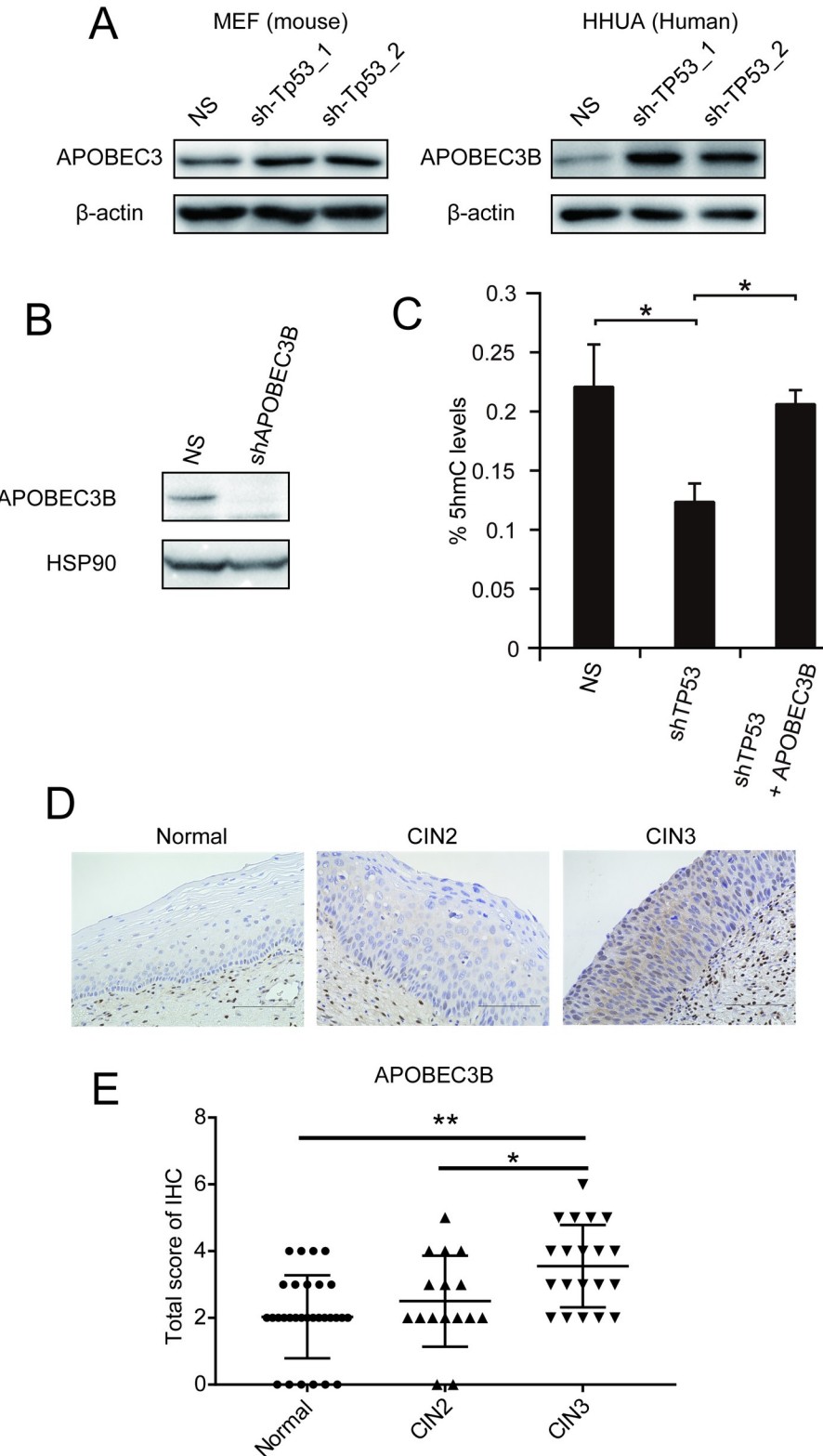

**Fig 3. A3B was induced during the transition from CIN2 to CIN3 in cervical tumors.** (A) Western blot analysis of APOBEC3 in sh-*Tp53* and sh-control cells. (B) Western blot analysis of A3B in sh-*TP53* and sh-control cells. (C) Western blot analysis of A3B in sh-*A3B* and sh-control cells. HSP90 was the loading control. (D) Representative images of IHC staining of A3B. Scale bar, 100 μm. (E) Dot plot shows the IHC scores for A3B expression in normal

cervical epithelium, CIN 2 and 3. The score was significantly higher for CIN 3 than for normal cervical epithelium and CIN 2. Error bars indicate the standard deviation. *, P<0.05; **, P<0.01.

Although the probability that CIN2 progresses to cancer is lower than CIN3, it is difficult to diagnose them correctly. Thus, evaluation of 5hmC status could be useful because 5hmC levels change markedly between CIN2 and CIN3. Additionally, given that 5hmC loss could reflect the mutation frequency, detection of 5hmC reduction might lead to precise identification of cases that are highly predisposed to cervical cancer.

In contrast to our findings, Su et al. reported that 5hmC expression levels increased from low-grade squamous intraepithelial lesions (LSIL) to high-grade squamous intraepithelial lesions (HSIL). In their study, the 5hmC levels throughout the full spectrum of cervical lesions were measured. In our study, we scored only basal and undifferentiated cells, except for differentiated cells in order to assess carcinogenesis [39]. They divided CIN cases into LSIL and HISL, and we divided them into CIN1, CIN2 and CIN3. Those factors might lead to different results, while both groups reported 5hmC loss in SCC.

In our study, there are some limitations. For example, we assessed only MEFs as a model of normal cells and we used a human cancer cell line because we were not able to cultivate normal human cervical epithelial cells *in vitro*. We intend to improve the experimental methods and assess these human cells as a normal model in future research. Also, we could not evaluate the mutation frequencies in cervical tumors directly. It might be particularly useful to investigate the relationship between 5hmC loss and mutation frequencies in CINs as a prospective cohort study.

In conclusion, 5hmC levels were decreased between CIN2 and CIN3 through the TP53-A3B pathway. Since A3B could impair genome stability, 5hmC loss might increase the chances of accumulating mutations and supporting the progression from CIN3 to cervical cancer. Thus, 5hmC could become a useful molecular biomarker not only for CIN2 and CIN3 diagnosis, but also for prediction of CIN cases that are more predisposed to cervical cancer.

## Supporting information

**S1 Fig.** (A) Representative images of IHC staining of 5mC with DAPI. Scale bar, 100 μm. (B) Dot plot shows the IHC scores for the level of 5hmC in normal cervical epithelium, CIN1, CIN2, CIN3 and cancer. Error bars indicate the standard deviation.
(TIF)

**S1 Table. Antibodies used for immunohistochemistry, ELISA and Western blot analysis.**
(DOCX)

**S2 Table. Scoring system for IHC immune-histochemical staining.**
(DOCX)

**S3 Table. Target sequences of the shRNAs.**
(DOCX)

**S4 Table. Primer sequences used in qRT-PCR experiments.**
(DOCX)

**S5 Table. Statistical analysis of 5mC levels in normal cervical epithelia and cervical tumors.**
(DOCX)

**S1 Raw data.**
(TIF)

**S2 Raw data.**
(TIF)

**S3 Raw data.**
(TIF)

**S4 Raw data.**
(TIF)

## Acknowledgments

We are grateful to Sawako Adachi, Mayu Mizumoto, Yuko Endo, and the members of Research Support Center, Graduate School of Medical Science, Kyushu University for their technical support.

## Author Contributions

**Conceptualization:** Ichiro Onoyama.

**Data curation:** Masaya Kato, Ichiro Onoyama, Minoru Kawakami, Emiko Hori.

**Formal analysis:** Masaya Kato, Ichiro Onoyama, Minoru Kawakami.

**Funding acquisition:** Ichiro Onoyama.

**Investigation:** Masaya Kato, Sachiko Yoshida, Keiko Kawamura, Keisuke Kodama, Emiko Hori.

**Methodology:** Masaya Kato, Ichiro Onoyama, Emiko Hori, Lin Cui, Yumiko Matsumura.

**Project administration:** Ichiro Onoyama.

**Resources:** Emiko Hori, Hiroshi Yagi, Kazuo Asanoma.

**Software:** Ichiro Onoyama, Emiko Hori, Kazuo Asanoma.

**Supervision:** Hiroshi Yagi, Kazuo Asanoma, Hideaki Yahata, Atsuo Itakura, Satoru Takeda, Kiyoko Kato.

**Validation:** Masaya Kato, Sachiko Yoshida, Keiko Kawamura, Keisuke Kodama, Yumiko Matsumura.

**Visualization:** Masaya Kato.

**Writing – original draft:** Masaya Kato.

**Writing – review & editing:** Ichiro Onoyama.

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
