## [Decision Letter · Decision Letter 0]

18 Feb 2020

PONE-D-20-00415

Downregulation of 5-hydroxymethylcytosine is associated with the progression of cervical intraepithelial neoplasia

PLOS ONE

Dear Dr Onoyama,

Thank you for submitting your manuscript to PLOS ONE. After careful consideration, we feel that it has merit but does not fully meet PLOS ONE’s publication criteria as it currently stands. Therefore, we invite you to submit a revised version of the manuscript that addresses all the points raised by the reviewers.

We would appreciate receiving your revised manuscript by Apr 03 2020 11:59PM. To enhance the reproducibility of your results, we recommend that if applicable you deposit your laboratory protocols in protocols.io, where a protocol can be assigned its own identifier (DOI) such that it can be cited independently in the future. For instructions see: http://journals.plos.org/plosone/s/submission-guidelines#loc-laboratory-protocols

We look forward to receiving your revised manuscript.

Kind regards,

Qian Tao

Academic Editor

PLOS ONE

Journal Requirements:

1. We noticed you have some minor occurrence(s) of overlapping text with the following previous publication(s), which needs to be addressed:

https://doi.org/10.1111/cas.13026

In your revision ensure you cite all your sources (including your own works), and quote or rephrase any duplicated text outside the Methods section. Further consideration is dependent on these concerns being addressed.

2. In your Methods section, please give the sources of any cell lines used in your study (HEK293).

3.

PLOS ONE now requires that authors provide the original uncropped and unadjusted images underlying all blot or gel results reported in a submission’s figures or Supporting Information files. This policy and the journal’s other requirements for blot/gel reporting and figure preparation are described in detail at https://journals.plos.org/plosone/s/figures#loc-blot-and-gel-reporting-requirements and https://journals.plos.org/plosone/s/figures#loc-preparing-figures-from-image-files. When you submit your revised manuscript, please ensure that your figures adhere fully to these guidelines and provide the original underlying images for all blot or gel data reported in your submission. See the following link for instructions on providing the original image data: https://journals.plos.org/plosone/s/figures#loc-original-images-for-blots-and-gels.

Reviewers' comments:

Reviewer's Responses to Questions

**Comments to the Author**

1. Is the manuscript technically sound, and do the data support the conclusions?

Reviewer #1: Yes

Reviewer #2: Partly

2. Has the statistical analysis been performed appropriately and rigorously? 

Reviewer #1: Yes

Reviewer #2: Yes

3. Have the authors made all data underlying the findings in their manuscript fully available?

Reviewer #1: Yes

Reviewer #2: No

4. Is the manuscript presented in an intelligible fashion and written in standard English?

Reviewer #1: No

Reviewer #2: Yes

5. Review Comments to the Author

Reviewer #1: The presence of methylation status 5mC and intermediate between methylation and unmethylation status 5hmC were examined, to correlate with tumor suppressor TP53 regulated activities and prognosis of cervical carcinoma. Insightful data has been described in the present MS, problems , however, were raised in term of presenting the work:

1.The paragraph under the title “Transfection and Lentivirus packaging “ in the part M&M (lines 11-17, p.13): What was transfected to the packaging cells HEK293T? The recombinant viruses was isolated from culture supernatant, but why not purified from cell lysates?

2.“The gels were transferred to nitrocellulose membranes”(Lines 3, 4, p.15): Separated proteins on the gels were transferred, not the gels;

3.“5% milk/TBST”: (Line 5, p.15): Was it defatted milk?

4.“Fig. 1. The levels of 5hmC and 5mC were reduced ...”:(lines 15-18, p.16): It is suggested that the figure legends and tables (the next pages) be placed separately after the main text.

5.“Table S2. Sequences used for qRT-PCR.”(lines 1, 2, p.24): Are the oligonucleotides listed PCR prmiers?

6.“For 5hmC and 5mC”(Line 7, p.12): It should be read “for detection of 5hmC and 5mC”.

Reviewer #2: Kato et al reported downregulation of 5-hydroxymethylcytosine (5hmC) and its association with progression of cervical intraepithelial neoplasia, as well as its possible mechanisms. The authors found that 5hmC levels were significantly decreased in cervical intraepithelial neoplasia 2/3 (CIN2/3) and tumor samples. The authors also found that knock-down p53 could decrease 5hmC and 5mC levels, while induce protein levels of DNMTs and TETs. APOBEC3B (A3B) expression was increased in p53-knockdown cells, as well as in CIN3 cases compared to normal epithelium and CIN2 cases. This work is interesting, however, there are some serious concerns which need to further clarification.

Comments:

1. Recently, another study reported that TET1 and 5hmC expression levels increases from normal cervix to Low- grade squamous intraepithelial lesion (LSIL), maximizes in high-grade squamous intraepithelial lesion (HSIL), and decreases in invasive cancer (P.-H. Su, et al. Cancer Letters 450 (2019) 53-62). The conclusion is inconsistent with the current study.

2. Some data in the present study are also contradicted. For example, in Figure 2, the expression levels of TETs and DNMTs were increased in p53-silencing MEF cells, while both 5hmC and 5mC levels were decreased. Generally, TETs and DNMTs expression levels should be consistent with 5hmC and 5mC levels.

3. The authors found that p53 may regulate 5hmC/5mC levels as well as TETs/DNMTs expression levels. These data should be repeated and further confirmed. Whether there is other mechanisms mediating p53 regulation on 5hmC and TETs.

3. The authors also studied the A3B expression and its association with CIN cases and p53. However, the authors didn’t establish any relationship between A3B and 5hmC.

6. PLOS authors have the option to publish the peer review history of their article (what does this mean?). If published, this will include your full peer review and any attached files.

Reviewer #1: No

Reviewer #2: No

---

## [Author Response · Author response to Decision Letter 0]

22 Sep 2020

Response to Reviewers

1. We noticed you have some minor occurrence(s) of overlapping text with the following previous publication(s), which needs to be addressed:

https://doi.org/10.1111/cas.13026

In your revision ensure you cite all your sources (including your own works), and quote or rephrase any duplicated text outside the Methods section. Further consideration is dependent on these concerns being addressed.

Response: 

We apologize for having repeated previous passages of text. We have revised our manuscript by rephrasing the duplicated text (page 5, lines 15- page 9, lines 13).

2. In your Methods section, please give the sources of any cell lines used in your study (HEK293).

Response:

We apologize for the omission. We have added the information regarding cell lines to the Methods section (page 7, lines 3-5).

3.PLOS ONE now requires that authors provide the original uncropped and unadjusted images underlying all blot or gel results reported in a submission’s figures or Supporting Information files. This policy and the journal’s other requirements for blot/gel reporting and figure preparation are described in detail at https://journals.plos.org/plosone/s/figures#loc-blot-and-gel-reporting-requirements and https://journals.plos.org/plosone/s/figures#loc-preparing-figures-from-image-files. When you submit your revised manuscript, please ensure that your figures adhere fully to these guidelines and provide the original underlying images for all blot or gel data reported in your submission. See the following link for instructions on providing the original image data: https://journals.plos.org/plosone/s/figures#loc-original-images-for-blots-and-gels.

Response:

We have provided the original uncropped and unadjusted images underlying all blot results. Please check supporting data. 

Response:

We have conformed to your recommendations. All of the data from this study are available upon request and we have uploaded the minimal anonymized data set in the Supporting Information files.

Response:

We have ensured that our information was updated.

 

Response to Reviewer #1

We thank the reviewer for the careful reading of our manuscript and for the statement that “insightful data has been described.” We also thank the reviewer for suggestions that we feel have helped us to improve our manuscript. Our specific responses to the points raised are as follows:

1.The paragraph under the title “Transfection and Lentivirus packaging “in the part M&M (lines 11-17, p.13): What was transfected to the packaging cells HEK293T? The recombinant viruses was isolated from culture supernatant, but why not purified from cell lysates?

Response:

We appreciate the thorough review of our manuscript. We used HEK293T cells as a packaging cell line for lentivirus production. We have now clarified this point in the Materials and Methods section of the revised manuscript (page 7, lines 3-5,). We isolated lentivirus according to the manufacturer’s instructions from Addgene. We have also clarified this point in the revised manuscript (page7, lines 11-16).

2.“The gels were transferred to nitrocellulose membranes”(Lines 3, 4, p.15): Separated proteins on the gels were transferred, not the gels;

Response:

As suggested by reviewer, we have now stated that separated proteins on the gels were transferred to nitrocellulose membranes (page 8, line 18- page 9, line 1).

3.“5% milk/TBST”: (Line 5, p.15): Was it defatted milk?

Response:

Yes, it was. I have revised the term to skim milk (page 9, line 2).

4.“Fig. 1. The levels of 5hmC and 5mC were reduced ...”:(lines 15-18, p.16): It is suggested that the figure legends and tables (the next pages) be placed separately after the main text.

Response:

We followed the Author guideline that says “Figure captions must be inserted in the text of the manuscript, immediately following the paragraph in which the figure is first cited.” and “Place each table in your manuscript file directly after the paragraph in which it is first cited.” 

5.“Table S2. Sequences used for qRT-PCR.” (lines 1, 2, p.24): Are the oligonucleotides listed PCR prmiers?

Response:

Yes, they are. We have revised the title of the Table (new Table S4).

6.“For 5hmC and 5mC” (Line 7, p.12): It should be read “for detection of 5hmC and 5mC”.

Response:

We appreciate your thorough review of our manuscript and for this comment. I have revised that (page 6, lines 6).

 

Response to Reviewer #2

Reviewer #2: Kato et al reported downregulation of 5-hydroxymethylcytosine (5hmC) and its association with progression of cervical intraepithelial neoplasia, as well as its possible mechanisms. The authors found that 5hmC levels were significantly decreased in cervical intraepithelial neoplasia 2/3 (CIN2/3) and tumor samples. The authors also found that knock-down p53 could decrease 5hmC and 5mC levels, while induce protein levels of DNMTs and TETs. APOBEC3B (A3B) expression was increased in p53-knockdown cells, as well as in CIN3 cases compared to normal epithelium and CIN2 cases. This work is interesting, however, there are some serious concerns which need to further clarification.

1. Recently, another study reported that TET1 and 5hmC expression levels increases from normal cervix to Low- grade squamous intraepithelial lesion (LSIL), maximizes in high-grade squamous intraepithelial lesion (HSIL), and decreases in invasive cancer (P.-H. Su, et al. Cancer Letters 450 (2019) 53-62). The conclusion is inconsistent with the current study.

Response:

We greatly appreciate the thorough review of our manuscript. This comment was highly insightful and helped us to improve the quality of our manuscript. As you pointed out, Su et al. reported that 5hmC expression levels increase from LSIL to HSIL. It appears that their scores were from whole epithelium, including differentiated cells and basal undifferentiated cells. On the other hand, we scored only the basal and undifferentiated cells, except for differentiated cells. In the case of assessing carcinogenesis, we believe that only the undifferentiated cells should be evaluated. Therefore, the results of their IHC are not totally inconsistent with ours. We have addressed this point in the revised manuscript (page 17, lines 10-16).

2. Some data in the present study are also contradicted. For example, in Figure 2, the expression levels of TETs and DNMTs were increased in p53-silencing MEF cells, while both 5hmC and 5mC levels were decreased. Generally, TETs and DNMTs expression levels should be consistent with 5hmC and 5mC levels.

Response:

As you pointed out, TETs and DNMTs increase 5hmC and 5mC. We examined the Western blot and TET1 and DNMT1 were strongly expressed in TP53-silenced MEFs. However, note that DNMT1 induces site-specific methylation of gene promoter regions that constitute very small portions of the whole genome. The specificity of DNMT1 might account for the low level of DNA methylation despite DNMT1 induction.

TETs promote DNA demethylation from 5mC to cytosine through 5hmC, 5-formylcytosine and 5-carboxylcytosine. When we knocked down TP53, the DNA methylation level was already low, as shown in Figure 2C. From those data, we hypothesized that DNA demethylation after TP53 knockdown might proceed and be completed very quickly. For that reason, we could not detect 5hmC upregulation after DNA demethylation was completed. We have clarified this point in the Discussion section of the revised manuscript (page 16, lines 4-12).

3. The authors found that p53 may regulate 5hmC/5mC levels as well as TETs/DNMTs expression levels. These data should be repeated and further confirmed. Whether there is other mechanisms mediating p53 regulation on 5hmC and TETs.

Response:

As you suggested, we reconfirmed that Tp53 knockdown led to 5hmC and 5mC reduction with a 2nd Tp53 shRNA as shown in Figure 2B and C. Also, we performed TP53 and APOBEC3B double knockdown experiments, and they showed that the TP53 and APOBEC3 pathway was involved in 5hmC regulation (new Figure 3C), although there might be other mechanisms that regulate 5hmC and TETs through TP53. We have now clarified this point in the Discussion section of the revised manuscript (page 16, line 13- page 17, line 2).

4. The authors also studied the A3B expression and its association with CIN cases and p53. However, the authors didn’t establish any relationship between A3B and 5hmC.

Response:

We apologize for the inadequate information regarding A3B and TP53. It was reported that tumor-associated TP53 mutants can promote A3B expression (Menendez D, et al. Mol Cancer Res; 15(6) 2017). As you suggested, we examined whether the TP-53 and A3B pathways were involved in 5hmC regulation using the HHUA cell line, which has a wild-type TP53. First, we confirmed that TP53 knockdown induced A3B expression (Figure 3A). Second, the 5hmC level was decreased after the knockdown of TP53 cells, and this 5hmC reduction was rescued by additional APOBEC3B knockdown. These data indicated that the TP53 and APOBEC3 pathway is, at least partly, involved in 5hmC regulation (new Figure 3C). We have now clarified this point in the Results and Discussion section of the revised manuscript (page 16, line 13- page 17, line 2).

---

## [Decision Letter · Decision Letter 1]

16 Oct 2020

Downregulation of 5-hydroxymethylcytosine is associated with the progression of cervical intraepithelial neoplasia

PONE-D-20-00415R1

Dear Dr. Onoyama,

We’re pleased to inform you that your manuscript has been judged scientifically suitable for publication and will be formally accepted for publication once it meets all outstanding technical requirements.

Kind regards,

Qian Tao

Academic Editor

PLOS ONE

Additional Editor Comments (optional):

Reviewers' comments:

Reviewer's Responses to Questions

**Comments to the Author**

1. If the authors have adequately addressed your comments raised in a previous round of review and you feel that this manuscript is now acceptable for publication, you may indicate that here to bypass the “Comments to the Author” section, enter your conflict of interest statement in the “Confidential to Editor” section, and submit your "Accept" recommendation.

Reviewer #1: All comments have been addressed

Reviewer #2: All comments have been addressed

2. Is the manuscript technically sound, and do the data support the conclusions?

Reviewer #1: Yes

Reviewer #2: Yes

3. Has the statistical analysis been performed appropriately and rigorously? 

Reviewer #1: Yes

Reviewer #2: Yes

4. Have the authors made all data underlying the findings in their manuscript fully available?

Reviewer #1: Yes

Reviewer #2: Yes

5. Is the manuscript presented in an intelligible fashion and written in standard English?

Reviewer #1: Yes

Reviewer #2: Yes

6. Review Comments to the Author

Reviewer #1: The returned MS has been carefully revised, and the questions raised by external peer reviewers have been adequately replied. The quality of the MS has been improved to level of being published. Acceptance is therefore recommended.

Reviewer #2: (No Response)

7. PLOS authors have the option to publish the peer review history of their article (what does this mean?). If published, this will include your full peer review and any attached files.

Reviewer #1: No

Reviewer #2: No

---

## [Editor Report · Acceptance letter]

23 Oct 2020

PONE-D-20-00415R1 

Downregulation of 5-hydroxymethylcytosine is associated with the progression of cervical intraepithelial neoplasia 

Dear Dr. Onoyama:

I'm pleased to inform you that your manuscript has been deemed suitable for publication in PLOS ONE. Congratulations! Your manuscript is now with our production department. 

Kind regards, 

on behalf of

Dr. Qian Tao 

Academic Editor

PLOS ONE